# Seipin Deficiency as a Model of Severe Adipocyte Dysfunction: Lessons from Rodent Models and Teaching for Human Disease

**DOI:** 10.3390/ijms23020740

**Published:** 2022-01-11

**Authors:** Jocelyne Magré, Xavier Prieur

**Affiliations:** Centre National de la Recherche Scientifique, Institut National de la Santé et de la Recherche Médicale, L’institut du Thorax, Université de Nantes, F-44000 Nantes, France; jocelyne.magre@univ-nantes.fr

**Keywords:** adipocyte, lipodystrophy, lipotoxicity, insulin resistance, seipin, adipocyte dysfunction, diabetic complications

## Abstract

Obesity prevalence is increasing worldwide, leading to cardiometabolic morbidities. Adipocyte dysfunction, impairing white adipose tissue (WAT) expandability and metabolic flexibility, is central in the development of obesity-related metabolic complications. Rare syndromes of lipodystrophy characterized by an extreme paucity of functional adipose tissue should be considered as primary adipocyte dysfunction diseases. Berardinelli-Seip congenital lipodystrophy (BSCL) is the most severe form with a near absence of WAT associated with cardiometabolic complications such as insulin resistance, liver steatosis, dyslipidemia, and cardiomyopathy. Twenty years ago, mutations in the *BSCL2* gene have been identified as the cause of BSCL in human. *BSCL2* encodes seipin, an endoplasmic reticulum (ER) anchored protein whose function was unknown back then. Studies of seipin knockout mice or rats demonstrated how seipin deficiency leads to severe lipodystrophy and to cardiometabolic complications. At the cellular levels, seipin is organized in multimers that are particularly enriched at ER/lipid droplet and ER/mitochondria contact sites. Seipin deficiency impairs both adipocyte differentiation and mature adipocyte maintenance. Experiments using adipose tissue transplantation in seipin knockout mice and tissue-specific deletion of seipin have provided a large body of evidence that liver steatosis, cardiomyopathy, and renal injury, classical diabetic complications, are all consequences of lipodystrophy. Rare adipocyte dysfunctions such as in BSCL are the key paradigm to unravel the pathways that control adipocyte homeostasis. The knowledge gathered through the study of these pathologies may bring new strategies to maintain and improve adipose tissue expandability.

## 1. Introduction

Obesity prevalence increases worldwide, leading to cardiometabolic morbidities [1]. The adipose tissue expandability hypothesis suggests that when adipose tissue reaches its personal threshold of expansion, additional lipids cannot be stored properly in adipocytes [2]. Instead, lipids deposit ectopically in various organs such as the liver, muscle, pancreas, heart, and vessels, where they activate lipotoxic pathways that compromise the function of these organs and contribute to insulin resistance [3]. Growing evidence supports that adipocyte dysfunction, which impairs adipose tissue expandability and plasticity, is central in the development of obesity-related metabolic complications [4]. Genetic studies revealed the existence of an allelic signature associated with a favorable adiposity pattern: high subcutaneous storage capacity, even associated with elevated BMI, lowers the risk of T2D and heart disease [5].

Conversely, rare syndromes of lipodystrophy characterized by an extreme paucity of functional adipose tissue are associated with severe cardiometabolic complications [6,7]. These are primary adipocyte dysfunction disorders, and the most severe is congenital generalized lipodystrophy (CGL), which is also named Berardinelli-Seip congenital lipodystrophy (BSCL) [8]. BSCL patients display a near absence of adipose tissue from birth or early infancy, affecting all metabolically fat pads (subcutaneous and visceral fat) and the early development of metabolic complications. Patients are recognized due to the lack of body fat, the visible veins, and prominent muscularity that cause a severe and striking phenotype. Patients with BSCL develop insulin resistance and associated cardiometabolic abnormalities soon after birth or in childhood. These include hyperinsulinemia and glucose intolerance to overt diabetes hepato-steatosis, cardiomyopathy, and hypertriglyceridemia [8]. This disease was reported in the 1950s in Brazil by Dr. Waldemar Berardinelli [9] and in Norway by Dr. Martin Seip [10]. Since then, ≈300/500 patients with BSCL have been reported in the literature with an estimated prevalence of ≈0.1 to 5 persons per million depending on the population.

Four subtypes are currently distinguished according to the causative gene involved—BSCL1 due to mutations in the gene encoding the enzyme 1-acylglycerol-3-phosphate O-acyltransferase 2 (AGPAT2) [11], BSCL2 due to mutations in the gene encoding seipin [12], BSCL3 and BSCL4 due to mutations in caveolae-associated proteins, respectively caveolin-1 (CAV1) [13] and PTRF (Polymerase 1 and Transcription Release Factor), also named CAVIN1 [14]. The first two subtypes linked to AGPAT2 or Seipin are, by far, the most frequent, representing about 95% of the patients [15,16,17].

Phenotypic differences have been observed among the known BSCL genotypes with those carrying *BSCL2* alterations displaying the most extreme phenotype. These patients have almost no detectable adipose tissue in any depots. In addition to the lack of the metabolically active adipose tissue, they lack the mechanical fat usually present in retro-orbital, palm, sole, and periarticular regions. They are at much higher risk for hypertrophic cardiomyopathy and have metabolic complications that are particularly severe with an earlier onset of diabetes [18,19]. BSCL2 patients often present with mild mental retardation and in a few cases with progressive encephalopathy with a poor prognostic during childhood [19,20,21].

The mechanism of adipose tissue loss is specific to each BSCL subtype depending on the biological function of the mutated gene [22]. Twenty years ago, when BSCL2 has been identified, seipin function was unknown. In silico studies could not predict any functional domain in seipin sequence [12]. Cellular studies revealed that seipin is an ER-anchored protein [23,24] and that seipin deficiency alters the lipid droplets (LD) morphology [23,24,25,26]. Seipin consists of a conserved loop located in the lumen of the ER, two transmembrane domains (amino acids 27–47 and 247–270), and the N- and C-terminal ends exposed in the cytosol [27,28]. Biochemistry studies [29], atomic force microscopy, and recently cryo-electron microscopy [30,31] have shown that seipin forms oligomers, either dodecamers or undecamers depending on species and cell type, to form a ringlike/donut structure of homo-oligomers with the transmembrane domains at the periphery and hydrophobic helices positioned at the inner surface of the ring, protruding into the luminal leaflet of the ER bilayer [30,31,32]. The luminal domain of each Seipin monomer adopts a similar fold with an eight-strand β-sandwich typical of certain lipid-binding proteins [30]. Recently, it was further shown that the hydrophobic helices contain serine residues that interact directly with TAG within the membrane. They concentrate TAG molecules, thereby facilitating lens formation and LD budding [33,34].

In addition, seipin is needed to ensure full ability of pre-adipocytes to differentiate into mature adipocytes [35,36]. However, it was unknown how seipin deficiency could lead to lipodystrophy, and more generally, how lipodystrophy leads to cardiometabolic complications was poorly understood. Thus, animal models of these conditions are very useful tools to study the function of the genes involved, to characterize the pathophysiology of these extreme cases of primary adipocyte dysfunction, and to test promising therapies. In this review, starting from the adipose tissue phenotype to the different cardiometabolic complications, we will discuss how the use of different rodent models, with total and tissue-specific deletions of seipin, largely increased our knowledge of the pathophysiology of BSCL.

## 2. Seipin Deficiency Leads to Severe Lipodystrophy

### 2.1. Adipose Tissue Loss

The total deletion of seipin in mice (seipin knockout, SKO, mice) leads to a dramatic loss of white adipose tissue (WAT) assessed by magnetic resonance imaging (MRI) and X-ray analysis and confirmed by dissection [37,38,39] in 8 to 12-week-old males and females. The residual adipose tissue represents less than 10% of the adipose tissue measured in WT littermates for the inguinal and mesenteric WAT [38], but gonadal WAT was completely absent. Similarly, 20-week-old SKO rats are severely lipodystrophic and display 95% adipose tissue loss [40]. In all models, leptin levels were 40 to 30% decreased, and adiponectin plasma concentrations were dramatically reduced, reaching a 90% decrease in 8 to 12-week-old mice [37,38,39]. Despite the consistent decrease in leptin levels, food intake was increased modestly in some reports [38,39] but unchanged in the other publications [37]. The gene expression profiling of WAT showed a genuine alteration with a strong reduction in the mRNA levels of key adipocyte markers such as perilipin1 (*Plin1*), peroxisome proliferator-activated receptor gamma (*Pparg*), adipocyte protein 2 (*Ap2*), and hormone-sensitive lipase (*Hsl*).

In young mice, we reported the presence of residual gonadal adipose tissue, and we described a loss of all WAT fat pads from 4 to 14-week-old mice along with a reduction in adiponectin levels [41]. This observation suggested that lipodystrophy might arise at least partially from a loss of mature adipocytes. Consistently, under the adipocyte specific-promoter of the AdipoQ gene, CRE-mediated *Bscl2* deletion leads to early (6 to 12 weeks of age) and severe lipodystrophy with a massive loss of gonadal and subcutaneous WAT and a 80% decrease in circulating adiponectin [42]. Under the AP2 promoter [43], seipin deletion leads to a less severe and more progressive adipose tissue loss, but the efficiency of the AP2 CRE has been challenged [44] and might explain this mild lipodystrophic phenotype. Intriguingly, in both models, in contrast to adiponectin, leptin levels were not altered, suggesting that the remaining adipose tissue was sufficient to maintain a critical production of the satiety adipokine [42,43]. Food intake was not increased. Finally, tamoxifen-inducible seipin deletion under the ERT2-AdipoQ promoter also induces a progressive loss of adipose tissue, strongly supporting that seipin is required for mature adipocyte maintenance [45,46].

### 2.2. Adipogenesis Impairment

Whether lipodystrophy is a developmental issue or an impairment of mature adipocyte maintenance remains unknown. Early reports demonstrated that seipin deficiency in mesenchymal stem cell lines C3H10T1/2 [35], in 3T3-L1 pre-adipocytes [36], and in mouse embryonic fibroblasts (MEF) [38] prevents normal adipogenesis. Notably, the induction of PPARγ target genes mRNA levels during adipogenesis is impaired in the absence of seipin, and thiazolidinediones (TZD), the potent PPARγ agonists, improve the ability of seipin-deficient cells to differentiate into adipocytes in vitro [36,38]. This suggests that seipin deficiency somehow blunts PPARγ activation during adipogenesis. Importantly, in SKO [38] and adipocyte-deficient seipin mice [43], TZD treatment increases adipose tissue mass and improves the metabolic phenotype of these animals. A clinical case study even reported an improvement of HbA1C and the presence of little subcutaneous WAT in a BSCL2 patient after 5 years of pioglitazone treatment [47].

At the mechanistic level, how seipin might regulate adipogenesis and PPARγ activity has been extensively studied. Evidence from yeast highlighted that seipin deficiency leads to abnormal accumulation of phosphatidic acid (PA) [25]. It has been then suggested that PA accumulation could inhibit PPARγ activity [48]. Importantly, PA is increased in seipin deficient adipose tissue in vivo [43]. Using 3T3-L1 preadipocytes, Yang’s lab demonstrated that seipin interacts with the glycerol-3-phosphate acyltransferase 3 (GPAT3) and inhibits its activity (Figure 1). They further demonstrated that GPAT3 inhibition improves adipogenesis in SKO MEF [49]. Finally, in vivo, the double KO mice for seipin and GPAT3 display an improvement in WAT properties and systemic insulin sensitivity as well as a reduction in liver steatosis [50]. GPAT3 catalyzes the synthesis of lysophosphatidic acid (LPA) that is transformed into PA by the LPA-acyltransferase-ß, AGPAT2; then, Lipin1, a phosphatidate phosphatase enzyme, allows the synthesis of DAG. Rochford’s Lab [32,51] demonstrated that Seipin oligomer interacts with AGPAT2 and Lipin 1, and they proposed that these physical interactions might be particularly relevant during adipocyte differentiation as both Lipin1 [52] and AGPAT2 [53,54] are adipogenesis regulators (Figure 1). They further demonstrated that seipin negatively regulates PA levels [51]. They also confirmed the physical interactions between seipin and GPAT3 but did not confirm that GPAT3 inhibition restores the adipogenic properties of seipin deficient pre-adipocytes. Altogether, seipin oligomers physically scaffold three key enzymes of the glycerophospholipid and triglyceride synthetic pathway, and these interactions are relevant during adipogenesis, at least in vitro. Further studies would be needed to specifically unravel how and when seipin oligomer orchestrates TAG synthesis during the adipogenic process.

### 2.3. Accelerated Lipolysis

Studying the differentiation of MEF isolated from SKO embryos, Chen et al. were the first to highlight an increase in basal lipolysis. The glycerol and NEFA release were elevated at days 4 to 6 of differentiation, and this was associated with a chronic activation of the protein kinase A (PKA) pathway, resulting in an increased phosphorylation of HSL and PLIN1 [39]. The pharmacological inhibition of lipase activity increased TAG storage in SKO MEF, suggesting that elevated basal lipolysis was, at least partially, responsible for adipogenesis impairment. This elevated lipolysis during MEF differentiation has been reproduced in another report [38]. Ex vivo, in SKO WAT explants, Chen et al. reported an increase in basal lipolysis [39], whereas we observed a decrease in basal and stimulated lipolysis [41]. As in vitro, lipolysis activation was maximal at day 4 to 6 of differentiation and was similar to control cells at days 8; we could speculate that the phenomenon is transitory given that at a certain stage, the residual adipose tissue is nearly devoid of full adipocytes. Aiming to test the relevance of this observation in vivo, Chen’s lab crossed SKO mice with adipose triglyceride lipase (ATGL) KO mice and demonstrated that ATGL ablation rescues the lipodystrophic phenotype and the metabolic complications associated with seipin deficiency [55]. ATGL ablation also improves the adipocyte differentiation of seipin-deficient MEF in vitro [55]. Finally, inducible deletion of seipin in developing adipocytes leads to an elevated NEFA release in WAT explants, which is associated with chronic activation of the PKA pathway [45]. Together with the increased NEFA release, several studies reported an increase in FA oxidation and thermogenesis [45,56] in seipin-deficient WAT. Recently, using an inducible deletion of seipin, we reported that this induction of FA catabolism was transitory, suggesting that it may be an early event accounting for the loss of TAG store in seipin-deficient adipose tissue [46]. Altogether, several pieces of evidence support that abnormally elevated NEFA release and catabolism might contribute to the lipodystrophic phenotype of seipin-deficient mice.

### 2.4. The Lipid Droplet Safe Guard

Early studies in yeast have shown that seipin localizes at ER/LD contact sites [23] and that seipin deficiency alters LD morphology in yeast [23,24,25] and human cells [26]. Seipin is required for the normal rate of LD formation [57] and is necessary for converting nascent to mature LD [58]. It has been further reported that seipin acts as an ER-to-LD targeting protein during LD maturation, recruiting lipids and proteins in a specific, stepwise, and time-regulated manner [59]. Consistently, in BSCL2 patients’ fibroblasts, ER/LD contacts appear abnormal and LD show aberrant mobility. This implication of seipin in orchestrating the ER-to-LD recruitment of lipids and proteins was recently confirmed in yeast [60]. Importantly, when seipin is trapped at cytosolic side of the nuclear envelope, it promotes the formation of LD at this site specifically. Finally, it has been demonstrated that seipin interacts with TAG and DAG and helps their flow to the LD by maintaining functional ER–LD contacts and counteracting LD ripening [33,34,61]. Indeed, seipin seems to rearrange the ER membrane at specific ER domains favoring LD biogenesis and maturation [34]. Altogether, a large body of work performed in cellular systems supports that seipin oligomers organize the ER membrane subdomains to promote normal LD assembly and to prevent TAG ripening (see review [62], Figure 1A). Although this has not been formally demonstrated during adipogenesis, it is highly likely that the lack of seipin prevents normal LD in mature adipocytes. In addition, as seipin binds TAG and DAG and targets them to the LD, we may hypothesize that in the absence of seipin, TAG and DAG would be misaddressed and therefore could be more easily the target of lipases. In addition, as suggested by Rao and Goodman, small and abnormal LDs could be the substrate for lipophagy that would result in unstimulated lipolysis [62]. Therefore, the fundamental role of seipin in LD homeostasis might explain both the impaired adipogenesis and the accelerated lipolysis reported in seipin deficiency.

### 2.5. Other Mechanisms Contributing to the Adipocyte Loss

Nevertheless, the function of seipin in mature adipocytes remains incompletely understood, and several elements need to be clarified. In 3T3-L1 adipocytes, we have shown that seipin is enriched at ER/LD contact sites under lipid loading and alternatively localizes at the ER/mitochondria contact sites also named MAMs [46]. Seipin interacts with MAM-associated calcium regulators, and seipin deficiency alters the ER to mitochondria Ca^2+^ flux, and this reduction is associated with a decrease in ATP production that is known to be dependent on mitochondrial Ca^2+^ levels [46]. These data are consistent with the work performed by Huang’s lab that demonstrated in drosophila cells that seipin regulates ER Ca^2+^ handling [63,64]. Inducible-seipin deletion in mice leads to a progressive adipocyte loss associated with mitochondrial dysfunction. It has been previously shown that mitochondrial dysfunction can compromise adipocyte properties and finally cell survival [65]. Therefore, we can hypothesize that seipin function in controlling mitochondrial activity may explain, at least partially, the adipocyte loss caused by seipin deficiency (Figure 1B).

Finally, seipin deficiency in mature adipocytes induces the expression of ER stress markers including the pro-apoptotic C/EBP homologous protein (CHOP) transcription factor [43,46]. Therefore, ER stress might be one of the mechanisms involved in adipocyte loss. Seipin deficiency induces lipid remodeling, with an increase in several phospholipid species [43]. Changes in the PL ratio in the ER membrane might be an ER stress trigger [66]. Therefore, it would be useful to fractionate adipocytes and to assess the ER lipid composition. In addition, alterations in ER Ca^2+^ store and ATP deprivation might be other ER stress inducers. Further study should investigate if correcting the ER stress caused by seipin deficiency prevents the adipocyte loss.

### 2.6. What about Brown Adipose Tissue?

Whereas SKO mice or rats display at least a 90% decrease in WAT mass, the alteration of interscapular brow adipose tissue (BAT) mass is less severe, reaching a 30 to 55% reduction [37,38,39,40]. In vitro, seipin deficiency does not prevent brown adipocyte differentiation [41,67]. SKO mice can cope with cold acclimation, but they display a reduction in maximal thermogenic adaptation and are resistant to pro-thermogenic ß3 agonist treatment [41]. Specific seipin deficiency in BAT leads to an over-activation of the thermogenic/pro-catabolic program associated with BAT atrophy, mitochondrial dysfunction, and cell death [67,68]. Seipin deficiency in BAT leads to cold intolerance [68], supporting a cell-autonomous role for seipin in brown adipocytes.

## 3. Metabolic Phenotype and Diabetic Complications

### 3.1. Adipose Tissue Loss Leads to Severe Insulin Resistance

BSCL is defined in human patients by a nearly complete lack of adipose tissue associated with severe insulin resistance, which progresses to overt diabetes. SKO mice or rats are indeed severely lipodystrophic, and their metabolic phenotype has been studied [37,38,39,40]. All models display a 30 to 60% decrease in leptin levels associated with an increase in food intake [38,39]. Adiponectin levels are strongly affected [37,38,39], and this is associated with an impairment in glucose tolerance, insulin sensitivity, and a marked hyper-insulinemia. Hyperglycemia is visible in a random fed state only, not in the fasting condition [37,38,39,40]. Thus, SKO mice display metabolic inflexibility, and this is particularly severe during fasting, as they display a default in ketone bodies production, a drop in glycemia, and they are unable to maintain their body temperature [41].

Intriguingly, SKO mice are hypotriglyceridemic [37,38,39], but post-prandial TAG levels are elevated, supporting a default in lipid handling [39,69]. Our study showed that the low TAG levels might be due to an increase in TAG-rich lipoprotein uptake in the liver of SKO mice [38]. Of note, seipin-deficient rats are hypertriglyceridemic, suggesting that the rat could be a better model to study lipoproteins in the context of lipodystrophy [40].

In SKO mice, adipocyte-specific expression of seipin is sufficient to improve adipose tissue mass, leptin and adiponectin levels, insulin sensitivity, and glucose tolerance [70]. Consistently, adipose tissue transplantation improves post-prandial TAG, plasma leptin, glucose tolerance, insulin sensitivity, and insulin levels [69,71]. Leptin replacement also improves carbohydrate metabolism even though the effect is less marked than adipose tissue transplantation [71]. Taken together, these data support that lipodystrophy is the main trigger of carbohydrate metabolism alterations in SKO mice.

All models of adipocyte-specific deletion of seipin display lipodystrophy but variable metabolic responses. AP2-driven seipin deletion leads to insulin resistance and glucose intolerance in 6-month-old animals [43]. Inducible seipin deletion leads to glucose intolerance and insulin resistance 3 months after tamoxifen injection only [45,46] if tamoxifen is reinjected every 3 months, avoiding the new differentiation of WAT adipocytes. In contrast, seipin deletion under the constitutive adipoQ promoter does not lead to glucose intolerance [42], but a 48 h high-fat diet (HFD) feeding is sufficient to trigger glucose intolerance, suggesting that those mice are highly susceptible to metabolic complications [42]. Altogether, the lack of adipose tissue is the major trigger of the metabolic complications associated with BSCL, and adipocyte seipin largely accounts for the development of carbohydrate metabolism abnormalities. However, as the metabolic complications appear later on or require an HFD trigger, this raises the question of the cell-autonomous function of seipin in metabolic tissues.

### 3.2. Liver Steatosis Is Not Due to Seipin Loss in the Liver

SKO mice and rats display an important hepatomegaly characterized by a massive liver steatosis [37,38,39] (Figure 2). To our knowledge, no report described the pathological state of the liver, i.e., simple steatosis or NAFLD, including fibrosis and inflammation. Liver lipid analysis revealed a 3 to 4-fold increase in TAG and in DAG content [37,38,72]. DAG is well known to contribute to liver insulin resistance, and indeed, insulin signaling is blunted in SKO mice but improves by adipose tissue transplantation [69]. The re-expression of seipin in adipocytes only is sufficient to reduce liver steatosis in SKO mice [70], and consistently, the adipocyte-specific deletion of seipin leads to liver TAG accumulation [42,43]. Two mice models of restricted hepatic deletion of seipin have been generated, and their characterization did not reveal liver steatosis nor the associated metabolic complications [72,73]. Altogether, these data support that seipin deficiency in adipocytes leads to lipodystrophy that induces liver steatosis and liver insulin resistance. Animal and cellular studies support that there is no obvious cell-autonomous function for seipin in the hepatocyte. Therefore, seipin deficiency is a good model to study the effects of adipose tissue dysfunction on NAFLD development. Further studies should be performed to report more precisely the liver phenotype (fibrosis, inflammation) beyond lipid accumulation.

### 3.3. A Valuable Model for Diabetic Cardiomyopathy

Cardiomyopathy frequently occurs in BSCL2 patients and is characterized by concentric left ventricular hypertrophy (LVH) often associated with diastolic dysfunction but generally with preserved systolic function [74]. The phenotype is quite heterogeneous, and the cause is unclear. Most precisely, it is not clear whether this cardiomyopathy is associated with an ectopic lipid accumulation, as the results in the literature are contradictory [74,75,76]. SKO mice have been a useful model to characterize the cardiac phenotype associated with seipin deficiency. Hearts from SKO mice are heavier [55,77,78] (Figure 2), and this was associated with a transcriptomic signature of an induction of the IGF1R/PI3K pathway [55]. Echography and MRI highlighted an LVH in SKO mice [55,77,78]. At the functional levels, the main issue is a diastolic dysfunction assessed by a decrease in the ratio between early (E) and late (atrial-A) ventricular filling velocity (E/A ratio) [77], an increase in isovolumetric relaxation time [77,78], an alteration in the myocardial performance, and a modest but significant reduction in the ejection fraction. Altogether, this represents a valid model for diabetic cardiomyopathy. Lastly, transverse aortic constriction has a more severe effect in SKO mice [79]. However, the reports diverge on the potential cause for the cardiac dysfunction. We observed that the LVH is positively correlated with hyperglycemia, pointing out a role of glucose overload in the development of cardiac dysfunction. Thus, we assessed a chronic activation of the hexosamine pathway compatible with a glucotoxic mechanism [77]. Importantly, sodium-glucose co-transporter-2 (SGLT2) inhibitor treatment improves hyperglycemia and cardiac parameters suggesting that hyperglycemia resulting from insulin resistance is the main trigger of cardiac dysfunction in SKO mice. Of note, in this report, cardiomyopathy was detected in the absence of fibrosis, lipid abnormalities, or AGE accumulation. In contrast, another study proposed that seipin deficiency promotes excessive lipid catabolism in the heart that alters cardiac function [55]. The last report proposed that seipin deficiency induces neutrophil infiltration and severe fibrosis and an alteration in the phosphorylation of the contractile protein Titin that might change its functional properties [78]. Interestingly, these authors also showed that the heart-specific deletion of seipin does not lead to any cardiac dysfunction, supporting that cardiomyopathy is a consequence of lipodystrophy and the subsequent insulin resistance. Another group reported that in SKO mice, perivascular adipose tissue loses its anti-contractile properties [80]. This might also contribute to the diastolic dysfunction assessed in these animals. Altogether, SKO mice recapitulate the cardiomyopathy features and are unique tools to study the consequences of adipose tissue dysfunction on cardiac physiology. Importantly, lipid accumulation was not reported in any of these models, supporting that it was not the trigger of cardiac dysfunction.

### 3.4. Other Metabolic Complications

Kidney injury is a widely spread complication of diabetes. Lipodystrophic SKO mice display polyuria, increased albumin, creatinine, and ion excretion (Figure 2). These functional abnormalities were associated with an increase in glomerular and mesangial surface, along with lipid deposition, AGE accumulation, and hallmarks of oxidative stress, suggesting that lipotoxicity and glucotoxicty were both triggers of renal abnormalities. Of note, no sign of fibrosis or inflammation was detected. Importantly, leptin treatment and adipose tissue transplantation restore the physiological and morphological kidney properties, suggesting that kidney injury is a consequence of adipocyte dysfunction [71].

Muscle insulin resistance is a key contributor to chronic hyperglycemia in patients with type 2 diabetes. Insulin signaling appears to be normal [69] or slightly increased [81]. Muscles of SKO mice appear to accumulate less TAG and more glycogen, but the consequences on muscle physiology and metabolism are not clear yet. Seipin deletion in muscles does not alter muscle metabolism nor insulin signaling [81]. Therefore, seipin-deficient mice do not display major muscle insulin resistance nor lipid accumulation.

Intriguingly, whereas *Bscl2*^−/+^ heterozygous mice are not lipodystrophic nor insulin resistant, one study revealed a decrease in insulin secretion in isolated islet ß-cells and in vivo in these animals [82]. Of note, neither lipotoxic nor glucotoxic hallmarks were reported in SKO mice islets (Figure 2). However, this does not translate into glucose intolerance. Specific BSCL2 deletion in the ß-cell is needed to state whether seipin could play a cell-autonomous role in this key tissue for metabolic homeostasis. Given that calcium signaling plays a key role in glucose-stimulated insulin secretion in this cell type, studying the role of seipin in ER calcium handling would be particularly relevant in ß-cells.

Finally, the only effect of seipin deficiency that appears independent of lipodystrophy in rodent models is the effect on brain function. As BSCL2 patients display mild mental retardation, several teams analyzed the central nervous system function in seipin-deficient rodents. They reported deficits in learning and memory [40,83] and in motor coordination [84] involving different mechanisms. These deficits have been observed in SKO mice and in neuronal-specific deficient (nSKO) mice but not in adipose-specific SKO mice, which indicates that they are not a consequence of lipodystrophy and illustrates the cell-autonomous role of seipin in nervous system. Both behavior and motor deficits are triggered by the reduction in PPARγ activity and interestingly, PPARγ agonists TZD alleviate the physiological alterations described above [83,84].

## 4. Conclusions

Twenty years ago, mutations in *BSCL2* have been identified as the first cause for BSCL in human. Ten years ago, the first mouse model lacking seipin was published. In the last decade, an impressive number of studies intended to unravel how seipin deficiency leads to lipodystrophy and to cardiometabolic complications. At the cellular level, seipin is an ER anchored protein organized in multimers that are particularly enriched at ER/LD contact sites but also at ER/mitochondria contact sites. Seipin enrichment at one or the other ER contact site is nutritionally regulated. Seipin interacts with proteins involved in TAG synthesis, LD homeostasis, calcium handling, and mitochondrial function. Seipin deficiency impairs both adipocyte differentiation and mature adipocyte maintenance. The study of seipin function reinforced the importance of LD initiation–maintenance–expansion and illustrated how the homeostasis of this organelle is key in the adipocyte good health. Seipin also appears to have an unexpected role in Ca^2+^ exchange, and future work might help us to understand how Ca^2+^ signaling participates in adipocyte maintenance. Altogether, by studying one mysterious protein with no apparent functional domain, we are learning a lot about adipocyte biology.

Thus, seipin deficiency leads to severe lipodystrophy and cardiometabolic complications. Regarding diabetic cardiomyopathy, which remains a poorly understood complication, the data support that cardiac dysfunction takes place without lipid ectopic deposition, and one report supported a central harmful role of glucotoxicity. Therefore, SKO mice are a valuable model for studying diabetic cardiomyopathy pathophysiology and potentially to test new promising therapies.

Collectively, the studies discussed in this review support that liver steatosis, cardiomyopathy, and renal injury, classical diabetic complications, are all consequences of lipodystrophy. “It is not how fat you are, it is what you do with it that counts” [85], once said A Vidal-Puig. Seipin deficiency strikingly demonstrates that the lack of safe storage for lipids is a major threat for health. In contrast, adipose tissue transplantation and leptin treatment largely improve the SKO phenotype, demonstrating that increasing adipose tissue storage capacity deeply improves the metabolic health of these animals. Rare adipocyte dysfunctions such as in BSCL are key paradigm to unravel the pathways that control adipocyte good health. The knowledge gathered through the study of these pathologies may bring new strategies to maintain and improve adipose tissue expandability.

## Figures and Tables

**Figure 1 ijms-23-00740-f001:**
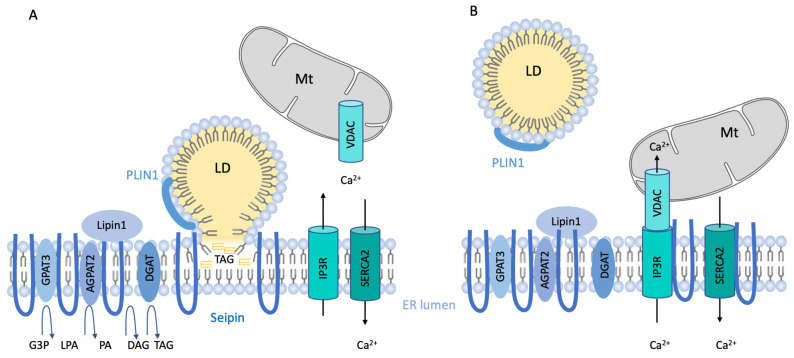
**Seipin function in mature adipocytes**. Seipin is an ER-anchored protein organized as oligomers. When cells are **lipid loaded** (**A**), seipin is enriched at the endoplasmic reticulum (ER)/lipid droplet (LD) contact sites and has been shown to be crucial in triglycerides (TAG) flow from the ER to the LD. Seipin interacts with Perilipin1 and with several TAG synthesis enzymes such as glycerol-3-phosphate acyltransferase (GPAT3), 1-acyl-sn-glycerol-3-phosphate acyltransferase beta (AGPAT2), and LIPIN. No interaction with diacylglycerol acyltransferases (DGAT) has been formally reported. (**B**)**-In the fasting** state, seipin is enriched at the ER/mitochondria (Mt) contact sites, also named mitochondria-associated membranes (MAM), and participates in ER/mitochondria calcium (Ca^2+^) flux and mitochondrial activity. Seipin is in close proximity to MAM Ca^2+^ regulators IP3(inositol 1,4,5-trisphosphate) receptor (IP3R), voltage-dependent anion channel (VDAC), and sarco-/ER Ca^2+^ ATPase 2 (SERCA2) as well as glycerol-3-phosphate (G3P), lysophosphatidic acid (LPA), phosphatidic acid (PA), and diacylglycerol (DAG).

**Figure 2 ijms-23-00740-f002:**
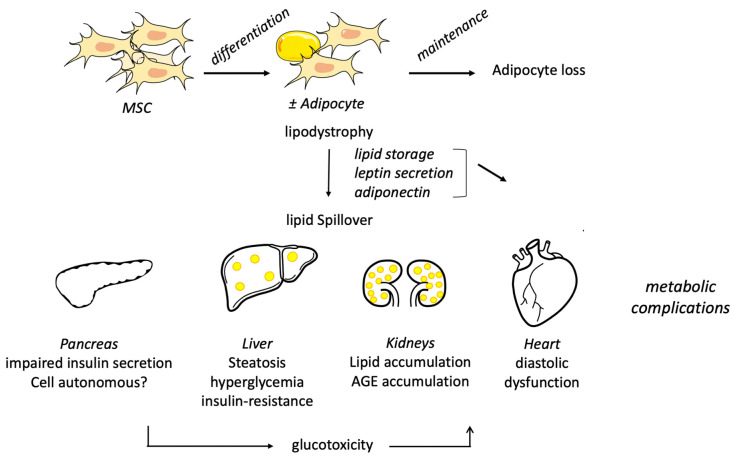
**Physiopathology in seipin-deficient mice.** Seipin deficiency leads to severe lipodystrophy that is the consequence of both adipogenesis impairment and alteration in mature adipocytes maintenance. The massive limitation in adipocyte storage promotes lipid ectopic accumulation in the liver and in the kidney. Liver steatosis leads to insulin resistance, which contributes to chronic hyperglycemia. Glucotoxic assault has been shown to contribute to kidney and heart dysfunction. Liver, kidney, and heart abnormalities are the consequences of adipose tissue dysfunction. On the other hand, pancreatic ß-cells display impairment in insulin secretion that has been attributed to a cell-autonomous function of seipin in this cell type. Mesenchymal stem cells (MSC), ±adipocytes: mix of adipocytes ± differentiated or apoptotic.

## Data Availability

Not applicable.

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
