# Peer review of "Seipin Deficiency as a Model of Severe Adipocyte Dysfunction: Lessons from Rodent Models and Teaching for Human Disease"

_ijms, 2022, doi:10.3390/ijms23020740_

Round 1
Reviewer 1 Report
In this study, authors reviewed about the effect of Seipin deficiency on lipid metabolism in animal models and roles of Seipin in the function of adipocyte. This is a well-reviewed paper taking on an important question.
Please correct notation of abbreviations in abstracts is required; endoplasmic reticulum (ER), lipid droplet (LD) and seipin knockout (SKO).
Please correct the following sentences in the abstract. "Rare adipocyte dysfunctions such as in BSCL are key paradigm to unravel the pathways that control adipocyte good health." In particular, the meaning of "adipocyte good health" is ambiguous. Wouldn't a better word to describe "normal metabolism of adipocytes" than that? Or replace it with a word that means normal function, etc.
figure1 : Please indicate TG and calcium in the figure.
figure 1 legend: in the word "aka MAM", what is the aka?
Author Response
R1 : In this study, authors reviewed about the effect of Seipin deficiency on lipid metabolism in animal models and roles of Seipin in the function of adipocyte. This is a well-reviewed paper taking on an important question.
Please correct notation of abbreviations in abstracts is required; endoplasmic reticulum (ER), lipid droplet (LD) and seipin knockout (SKO).
This has been corrected
Please correct the following sentences in the abstract. "Rare adipocyte dysfunctions such as in BSCL are key paradigm to unravel the pathways that control adipocyte good health." In particular, the meaning of "adipocyte good health" is ambiguous. Wouldn't a better word to describe "normal metabolism of adipocytes" than that? Or replace it with a word that means normal function, etc.
We have now replaced with “adipocyte homeostsasis”
figure1 : Please indicate TG and calcium in the figure.
This has been done
figure 1 legend: in the word "aka MAM", what is the aka?
"aka" is the abbreviation used for "also known as". However, to avoid any confusion, we have now replaced it with “also named MAM”
Reviewer 2 Report
The article prepared by Jocelyne Magre and Xavier Prieur fulfils a well-founded need to provide important data regarding metabolic disturbances, including insulin resistance, hypertriglyceridemia, and liver steatosis associated with Seipin deficiency. Seipin is an integral endoplasmic reticulum membrane protein encoded by BSCL2/Bscls2 gene, Berardinelli and Seipin congenital lipodystrophy (BSCL) being a rare autosomal genetic disease characterized by an almost complete lack of white adipose tissue.
The article is well-balanced and globally well written, such that the paper will be well received by peers in the field.
To improve the quality of the manuscript, we recommend:
- In addition to the subcutaneous and visceral fat tissue, lipids can also be stored in so-called ectopic fat. According to this aspect, in reference 3 (lines 32-34) there are no data regarding the vessels fat deposit (perivascular adipose tissue), so the authors should revise the information and add supplementary references (see Costa et al., 2018 DOI: 10.3389/fphys.2018.00235; Britton and Fox, 2011 doi.org/10.1161/CIRCULATIONAHA.111.077602);
- A brief report related to the Seipin structure in the Introduction section could be a good factor to complete the data presented;
- In line 54 the authors should add a reference related to the prevalence of BSCL;
- All the abbreviations used in Figure 1 should be explained in the legend;
- To avoid misinterpretation the authors should complete the name of the cell types presented in Figure 2;
- As minor comments - Bscl2 gene is written in different styles in the main text (see lines 57 and 87); some other abbreviations should be explained by the authors such as ER in line 74, MRI in line 88, mRNA in line 97, or Plin1, Pparg, aP2, and Hsl in line 98.
Author Response
The article prepared by Jocelyne Magre and Xavier Prieur fulfils a well-founded need to provide important data regarding metabolic disturbances, including insulin resistance, hypertriglyceridemia, and liver steatosis associated with Seipin deficiency. Seipin is an integral endoplasmic reticulum membrane protein encoded by BSCL2/Bscls2 gene, Berardinelli and Seipin congenital lipodystrophy (BSCL) being a rare autosomal genetic disease characterized by an almost complete lack of white adipose tissue.
The article is well-balanced and globally well written, such that the paper will be well received by peers in the field.
To improve the quality of the manuscript, we recommend:
In addition to the subcutaneous and visceral fat tissue, lipids can also be stored in so-called ectopic fat. According to this aspect, in reference 3 (lines 32-34) there are no data regarding the vessels fat deposit (perivascular adipose tissue), so the authors should revise the information and add supplementary references (see Costa et al., 2018 DOI: 10.3389/fphys.2018.00235; Britton and Fox, 2011 doi.org/10.1161/CIRCULATIONAHA.111.077602);
We are sorry but we do not understand what the reviewer is referring to as the author names and the DOI of the citation do not match and do not seem to refer to perivascular adipose tissue. Could the reviewer please re-formulate this suggestion?
A brief report related to the Seipin structure in the Introduction section could be a good factor to complete the data presented
This has been added as follow:
Cellular studies revealed that seipin is an ER anchored protein 1,2 and that seipin deficiency alters the lipid droplets (LD) morphology21-24. Seipin consists of a conserved loop located in the lumen of the ER, two transmembrane domains (amino acids 27-47 and 247-270) and the N- and C-terminal ends exposed in the cytosol6,7. Biochemistry studies 8, atomic force microscopy and recently cryo-electron microscopy 9,10 have shown that seipin forms oligomers, either dodecamers or undecamers depending on species and cell type, to form a ringlike/donut structure of homo-oligomers with the transmembrane domains at the periphery and hydrophobic helices positioned at the inner surface of the ring, protruding into the luminal leaflet of the ER bilayer 9-11 . The luminal domain of each seipin monomer adopts a similar fold with 8-strand β-sandwich typical of certain lipid binding proteins9. Recently, it was further shown that the hydrophobic helices contain serine residues that interact directly with TAG within the membrane. They concentrate TAG molecules thereby facilitating lens formation and LD budding 12,13.
In line 54 the authors should add a reference related to the prevalence of BSCL;
The sentence has been modified as follow:
The first two subtypes linked to AGPAT2 or Seipin are, by far, the most frequent representing about 95% of the patients15-17.
All the abbreviations used in Figure 1 should be explained in the legend
This has now been corrected as follow:
Figure 1: Seipin function in mature adipocytes. Seipin is an ER anchored protein organized as oligomers. When cells are lipid loaded (A), Seipin is enriched at the endoplasmic reticulum (ER)/lipid droplet (LD) contact sites and has been shown to be crucial in triglycerides (TAG) flow from the ER to the LD. Seipin interacts with Perilipin1 (PLIN1) and with several TAG synthesis enzymes such as glycerol-3-phosphate acyltransferase (GPAT3), 1-acyl-sn-glycerol-3-phosphate acyltransferase beta (AGPAT2) and LIPIN. No interaction with diacylglycerol acyltransferases (DGAT) has been formally reported. B-In the fasting state, seipin is enriched at the ER/mitochondria (Mt) contact sites, also named mitochondria-associated membranes (MAM), and participates to ER/mitochondria calcium (Ca2+) flux and mitochondrial activity. Seipin is in close proximity to MAM calcium regulators IP3(inositol 1,4,5-trisphosphate) receptor (IP3R), Voltage-dependent anion channel (VDAC) and sarco-/ER Ca2+ ATPase 2 (SERCA2). Glycerol-3-phosphate (G3P), lysophosphatidic acid (LPA), phosphatidic acid (PA), diacylglycerol (DAG).
To avoid misinterpretation the authors should complete the name of the cell types presented in Figure 2;
We have named the cells MSC (Mesenchymal stem cells) and ± Adipocytes
As minor comments - Bscl2 gene is written in different styles in the main text (see lines 57 and 87); some other abbreviations should be explained by the authors such as ER in line 74, MRI in line 88, mRNA in line 97, or Plin1, Pparg, aP2, and Hsl in line 98.
As concerns the different styles, we usually try as much as we can to respect the rules of nomenclature with:
- genes in italic for all species and for human genes, in upper case whereas for animal genes, the first letter in upper Case and the following ones in lower case
- proteins in upper case whatever the species. We have corrected Ap2 although people often write aP2. We have now explained the abbreviations